# Anatomy of Major Duodenal Papilla Influences ERCP Outcomes and Complication Rates: A Single Center Prospective Study

**DOI:** 10.3390/jcm9061637

**Published:** 2020-05-28

**Authors:** Gheorghe G. Balan, Mukul Arya, Adrian Catinean, Vasile Sandru, Mihaela Moscalu, Gabriel Constantinescu, Anca Trifan, Gabriela Stefanescu, Catalin Victor Sfarti

**Affiliations:** 1Faculty of Medicine, Grigore T. Popa University of Medicine and Pharmacy, 700115 Iasi, Romania; balan.gheo@me.com (G.G.B.); mmoscalu@yahoo.com (M.M.); ancatrifan@yahoo.com (A.T.); cvsfarti@gmail.com (C.V.S.); 2New York Presbitarian Brooklyn Methodist Hospital, New York, NY 11215, USA; mklarya@gmail.com; 3Faculty of Medicine, Iuliu Hatieganu University of Medicine and Pharmacy, 400012 Cluj-Napoca, Romania; catinean@gmail.com; 4Department of Gastroenterology, Clinical Emergency Hospital of Bucharest, 014461 Bucharest, Romania; 5Faculty of Medicine, Carol Davila University of Medicine and Pharmacy, 020021 Bucharest, Romania; gabrielconstantinescu63@gmail.com

**Keywords:** post-ERCP complications, cannulation, post-ERCP pancreatitis, morphology

## Abstract

Background: Endoscopic retrograde cholangiopancreatography (ERCP) has been one of the most intensely studied endoscopic procedures due to its overall high complication rates when compared to other digestive endoscopy procedures. The safety and outcome of such procedures have been linked to multiple procedure- or patient-related risk factors. The aim of our study is to evaluate whether the morphology of the major duodenal papilla influences the ERCP outcomes and complication rates. Methods: A total of 322 patients with a native papilla have been included in the study over an eight month period. Morphology of the papilla has been classified into normal papilla and four anatomical variations (Type I-IV). All patients have been prospectively monitored over a 15 day period after ERCP. Procedural outcomes and complication rates have been registered. Results: Morphology of the papilla influences both overall complication rates (95%CI, *p* = 0.0066) and post-ERCP pancreatitis rates (95%CI, *p* = 0.01001) in univariate analysis. Type IV papillae have proven to be independent risk factors for post-ERCP pancreatitis in multivariate analysis (OR = 12.176, 95%CI, *p* = 0.005). Type I papillae have been significantly linked to difficult cannulation (AUC = 0.591, 95%CI, *p* = 0.008); Conclusions: In the monitored cohort morphology of the major duodenal papilla has significantly influenced both ERCP outcomes and post-procedural complication rates.

## 1. Introduction

The first anatomical description of the major duodenal papilla was published during the late 19th century within the Hunterian Lectures [1] and it has since been a multidisciplinary issue debated by anatomists, radiologists, surgeons and gastroenterologists. For digestive endoscopists it has played an important role in developing techniques for endoscopic retrograde cholangiopancreatography (ERCP), as cannulation of the papilla is essential for the success of such procedures. Therefore, after more than forty years since the procedure has been developed, the outcomes and safety of ERCP are still an intensely debated issue. Acquiring the skills for a safe and successful bile duct cannulation is an essential step for the overall efficiency of the procedure [2,3], as difficulties in cannulation have often been linked to post-ERCP adverse events or poor outcomes [4,5]. Difficult cannulation has constantly been regarded as an independent risk factor for post-ERCP pancreatitis [6,7,8]. Furthermore, inability to cannulate the papilla leads to procedural failure and can require subsequent alternative techniques to be used [9].

Despite the essential role of deep duct cannulation in the procedural safety and success, to date there is only limited research concerning the impact of papillary morphology on bile duct cannulation [10,11,12]. On the other hand, any experienced ERCP endoscopist can differentiate multiple morphologies of the major papilla and their impact on choosing different bile duct cannulation strategies [13,14]. Thus, a well renowned research team from Scandinavia has developed in 2017 the first interobserver- and intraobserver-validated classification of the endoscopic appearance of the papilla, stressing its role in creating a common reporting system for endoscopists [15]. Recently, the same team has shown through a prospective multicenter study that the morphology of the major papilla affects bile duct cannulation and an anatomy-based approach should be included in ERCP training programs [16].

Given the deemed needed completion of the current validated classification of the papillary morphology, the aims of our study were to firstly expand upon the existing classification including anatomical variations of the papilla that would cover some of its missing morphologies. Secondly, we prospectively evaluated whether such expanded papillary types influence ERCP outcomes and complications rates in a cohort of consecutive patients.

## 2. Materials and Methods

### 2.1. Patients

We have prospectively monitored patients referred for therapeutic ERCP within the Institute of Gastroenterology and Hepatology of Iasi, an emergency-based tertiary center in Romania, between 1 January and 31 August 2018.

All patients were managed for both malignant and benign bile duct diseases. Inclusion criteria consisted of: (i) indication for ERCP: bile duct stones (including cholangitis and biliary pancreatitis), cholangiocarcinoma, pancreatic cancer and chronic pancreatitis, bile duct injuries, various extrinsic compressions, primary sclerosing cholangitis and post-liver transplantation strictures; (ii) a native papilla; (iii) age above 18 years; (iv) possibility for follow-up at 15 and 60 days; (v) expressed consent for inclusion in the study. Exclusion criteria were: (i) previous ERCP or sphincterotomy; (ii) presence of ampulary tumors or tumors invading the papilla; (iii) postoperative altered anatomy; (iv) duodenal and/or bilio-pancreatic trauma; (v) cannulation of the minor papilla; and (vi) impossibility for achieving either papilla classification or proper follow-up. Data regarding patient demographics and ERCP indication were recorded in all cases.

### 2.2. Classification of Papillary Morphology and Procedure Documentation

During all ERCP procedures the duodenoscope was advanced to the papillary region and morphology of the papilla was assessed. Its appearance was classified in either regular anatomy or one of the four anatomical variations. The classification expansion used within the study was based on a former definition of the papillary anatomical variations previously published by Canard et al. in 2011 [17]. By adapting the former description of anatomical variants, we obtained an objective and reproducible classification add-on suitable for our study design. A regular papilla was described as presenting the following features: frenulum, orifice, recessus and infundibulum (intraduodenal portion of the common bile duct) as shown in Figure 1. Anatomical variations were classified in: Type 1: small and/or retracted papilla, without a recessus and infundibulum; Type 2: papilla with a small infundibulum with no recessus and a poorly defined orifice; Type 3: papilla with a large protruding and/or pendulous infundibulum and visible orifice; Type 4: large papilla with multiple overlying folds over the orifice (commonly referred to as a *hooded* papilla or a *Shar-Pei dog* papilla). Graphical representations of the anatomical variations are illustrated in Figure 2. Presence of duodenal diverticula has been documented. If the endoscopic appearance of the papilla could not be classified, a second experienced endoscopist was involved and decision to either classify and include or exclude the patient has been taken within the team by consensus.

Video documentation of the ERCP procedure was performed in all cases. Bile duct cannulation was monitored. The number of intentional contacts with the papilla for attempted cannulation was noted. The time between the first intentional contact and bile duct cannulation confirmed by fluoroscopy was also measured. Unintentional guidewire passages in the main pancreatic duct were recorded. The need for freehand precut sphincterotomy, fistulotomy or transpancreatic septotomy was separately documented. Difficult cannulation was defined in accordance with the criteria established by the European Society for Gastrointestinal Endoscopy (ESGE) clinical guideline as: more than 5 contacts with the papilla while attempting to cannulate; more than 5 min attempting to cannulate following first intentional contact of the papilla; more than one unintended pancreatic duct cannulation or opacification [4].

ERCP procedures were performed by two experienced endoscopists (with an experience of over 1000 ERCPs performed) and two senior fellows in advanced endoscopy (with an experience of over 180 ERCPs performed). The guidewire-assisted technique was used for primary biliary cannulation. Freehand needle knife sphincterotomy (from the orifice), needle knife fistulotomy (above the orifice) were performed following the endoscopist’s decision only in the case of difficult standard cannulation. Transpancreatic biliary sphincterotomy has been considered after the second unintentional pancreatic duct cannulation. Mixed electrosurgical current was used in all cases for papillotomy. Complete biliary sphincterotomy and partial papillotomy were recorded. Not all patients received papillotomy.

All patients received preanesthetic assessment before ERCP. Administration of intrarectal indometacin or diclofenac and indication of the intravenous hydration protocol were approved by the anesthesia provider for each patient. Intrarectal indometacin or diclofenac has been administered to all patients prior to the procedures. All patients received hydration with lactated Ringer’s solution 3 mL/kg/h during ERCP. Patients with difficult cannulation and without easy pancreatic stenting (defined as repeated pancreatic duct cannulation and/or performance of transpancreatic biliary sphincterotomy) continued to receive lactated Ringer’s solution 20 mL/kg bolus after ERCP, followed by 3 mL/kg/h hydration for 8 h after ERCP. Patients with easy pancreatic stenting were inserted a 5 Fr, 5 cm prophylactic pancreatic stent and received no supplementary post-procedural hydration. No patient was managed by double guidewire cannulation technique as it is not included in the local ERCP protocol.

Data regarding the methods used for cannulation, cannulation attempts and time to successful cannulation, bile duct anatomy, insertion of stents, brush cytology or balloon sphincteroplasty were recorded for univariate analysis and multiple regression.

### 2.3. Prospective Evaluation

All patients were hospitalized for at least 24 h after ERCP. A thorough clinical assessment has been performed in all patients for at least 24 h after ERCP, including body temperature and vitals, as well as a biochemical panel including complete cell count, C reactive protein levels, lipase levels, liver and kidney function tests. Discharge was decided only by a consultant. Further hospitalization and monitoring of patients was performed depending on the presence of post-ERCP complication or on the overall recovery of patients. All patients were prospectively followed up by physical examination and assessment of symptoms at 15 and 60 days following the procedure in the ambulatory outpatient clinic. Adverse events including post-ERCP pancreatitis, bleeding, infection and perforation were defined according to consensus criteria of the American Society of Gastrointestinal Endoscopy (ASGE) [8]. Definition of post-ERCP pancreatitis was done after the revised Atlanta classification criteria [18]. The post-ERCP infections consisted of cholangitis, cholecystitis, liver abscess and symptomatic bacteremia episodes including the suspected duodenoscope-transmitted multidrug resistant infections. Post-procedural mortality was recorded.

### 2.4. Statistical Analysis

Statistical analysis and calculations were made with the SPSS version 25 (IBM, Armonk, NY, USA). Numerical data was reported as mean values and standard deviation. Comparisons within the analyzed groups were performed with the ANOVA and Kruskal–Wallis tests and displayed in box and whisker plots. The prediction power of variables was obtained using receiver operating characteristic (ROC) curves and by calculating the area under the curve (AUROC). Categorical variables were presented in absolute (*n*), or relative (%) numbers. The Pearson, M-L or Yates Chi-square test were used to obtain correlation r parameters. Equality of variances was assessed by using the Levene’s test.

We searched for possible predictors of each post-ERCP adverse event and of the cumulative complications rate. In addition to the papillary morphology, univariate analysis was also conducted for other procedure-related variables. Univariate analysis and multiple regression of predictors were performed after the logistic regression model. Odds ratio (OR) and Wald test were calculated. Marked effects of tests were considered significant at *p* < 0.05.

All patients gave their informed consent for inclusion before they participated in the study. The study was conducted in accordance with the Declaration of Helsinki, and the protocol was approved by the Research Ethics Committee of the Grigore T. Popa University of Medicine and Pharmacy of Iasi, Romania, Approval for Doctoral Research Series J, No. 0038034 issued for Gheorghe G. Balan.

## 3. Results

A total of 403 consecutive patients were referred for inclusion in the study. After applying inclusion and exclusion criteria, a total of 322 patients met the criteria and participated in the prospective study. Due to patient selection and model design, classification of the papilla into either regular morphology or one of the anatomical variants was possible in all cases. Distribution of the papilla types is shown in Table 1. As expected, a regular papilla was the most frequently encountered endoscopic feature, accounting for 52.1% of cases, followed by the Type 2 papilla found in 18.9% of cases.

The impact of papillary morphology on cannulation and the overall post-ERCP adverse events rates have been assessed. Subsequently, data regarding the predictive power of the papillary morphology on complication rates is shown in a multivariate regression and compared to that of other procedure-related variables. The overall post-ERCP 60 day survival rate was 95.4% in patients without post-ERCP adverse events, and 82.6% for patients where adverse events occurred.

### 3.1. Impact of Papillary Morphology on Cannulation

The overall frequency of difficult cannulation, regardless of the papilla types, was 34.4% (95%CI: 0.550–0.683). Within the different anatomical variations there were non-homogenous rates of difficult cannulation, as shown in Table 2. Nevertheless, the Type 1 small and retracted papillae have been significantly more difficult to cannulate (66.7%, r = 0.282, *p* = 0.00358) when compared to both regular papillae and other anatomical variations. As shown in Figure 3, there was a significant predictive power of the papillary aspect on the rate of difficult cannulation (AUC = 0591, 95%CI: 0.526–0.655, *p* = 0.008). Protruding and hooded papillae tended to be more difficult to cannulate (46.9% and 40%, r = 0.282), while regular papillae were the least associated with a difficult cannulation (25%, r = 0.282).

Given the overall significance and predictive power of papillary morphology on the rates of difficult cannulation, the next step was to assess the impact of different papilla types on the number of cannulation attempts and the total time for cannulation. As shown in Figure 4, the patients with Type 1 small and/or retracted papillae needed significantly more cannulation attempts when compared to those with a regular papilla (mean value of 5.5 attempts, *p* = 0.00108). However, the other anatomical variations did not seem to require more cannulation attempts than a regular papilla.

On the other hand, difficult cannulation rates were more clearly delineated by the time required until successful cannulation among the different types of papilla. Thus, as shown in Figure 5, Type 1 papillae needed significantly more time (mean value of 5.6 min) until successful deep cannulation was achieved compared to both regular papillae (mean value 3.7 min, *p* = 0.00181), and the other anatomical variations: Type 2 (mean value 3,8 min, *p* = 0.0014), Type 3 (mean value 3.8 min, *p* = 0.0008) or Type 4 (mean value 4.1 min, *p* = 0.0022).

The overall rates of failed deep cannulation reached 9% within the analyzed cohort with non-significant differences between the papilla types (r = 0.0247, *p* = 0.9096). Knowing that difficult cannulation is an individual risk factor for the post-ERCP adverse events rates, and that it has a significant correlation with the papilla’s appearance, our next approach was to evaluate whether the papillary morphology is associated with higher rates of post-procedural complications.

### 3.2. Impact of Papillary Morphology on the Overall Post-ERCP Adverse Events Rate. Univariate Analysis

As shown in Table 3, the appearance of the papilla major was significantly associated in univariate analysis to both the overall post-ERCP adverse events rates (*p* = 0.006) and the post-ERCP pancreatitis (PEP) rates (*p* = 0.01). No significant correlation was found for post-ERCP bleeding and infections. Perforations were scarce in the studied cohort, making statistical analysis impossible. There were only 2 Type 3 perforations both occurring in patients with regular papillae.

Patients with Type 4 large or hooded papillae had overall significantly higher post-ERCP adverse events rates (44% compared to 16.6% in patients with a regular papilla, 95%CI). The same was observed for PEP rates (28% in patients with Type 4 papillae, versus 10% in patients with a regular papilla, 95%CI). The frequency of PEP within the cohort regardless of the papilla type was 8.68%. In cases with difficult cannulation, the PEP rate increased significantly to 17.8% compared to only 4.91% when cannulation was not difficult (r = 0.614, *p* = 0.00007). Such results seem inconsistent, given the previously proven impact of Type 1 papillae on difficult cannulation.

We searched for a possible explanation and evaluated the use of rescue alternative papillotomy techniques for different types of papillae. Table 4 shows how the use of alternative access papillotomy techniques is distributed according to the endoscopic aspect of the papilla. Although not statistically significant, as shown in Table 4, patients with large papillae (Type 3 and 4) seem to require more frequently access papillotomy such as freehand precut or fistulotomy (r = 0.377, *p* = 0.014) compared to the patients with smaller types of papilla (regular and Type 1). Nevertheless, this does not thoroughly explain the former correlation.

We then searched the distribution of pancreatic stenting for each type of papilla. Interestingly, although not statistically significant, 8% of the patients with Type 4 papillae required pancreatic prophylactic stenting, compared to only 2.98% of patients with regular papillae (*p* = 0.754), as seen in Table 5. The lack of statistically significant data is due to a low number of patients with pancreatic stents. Nevertheless, there is a clear delineation that might contribute to the correlation between bulgy and hooded Type 4 papillae and the risk for overall post-ERCP adverse events and PEP. Given such a statistically significant correlation in univariate analysis, we proceeded with the multiple regression that would conform and increase reliability of the results.

### 3.3. Multiple Regression

In order to achieve a reliable multiple regression, besides papillary morphology, we also included other predictive procedure-related parameters that were significantly correlated with the post-ERCP adverse event rates following univariate analysis. Table 6 summarizes the multivariate correlations between procedure-related variables and the post-ERCP adverse events.

Results from the univariate analysis regarding the correlation between papillary morphology and the overall risk for post-ERCP complications have not been confirmed. Yet, a positive OR of 1.45 for Type 4 papillae shows a possible, although non-significant (*p* = 0.458, 95%CI: 0.540–3.932), correlation. On the other hand, difficult cannulation was found to be an individual and predictive risk factor for the overall post-ERCP risk of adverse events (OR 2.744, *p* = 0.0001, 95%CI: 1.526–4.933). As morphology of the papilla was significantly correlated with the rate of difficult cannulation, it can be assumed that such morphology could just have an indirect effect on the overall post-ERCP adverse events. The presence of duodenal diverticula was also considered a potential risk factor after univariate analysis, but no significant correlation has been proven after multivariate analysis.

Regarding the PEP, the presence of a Type 4 papilla (OR 12.176, *p* = 0.005, 95%CI: 2.131–69.567) was found to be an independent and predictive risk factor for PEP. Moreover, the presence of difficult cannulation (OR 2.775, *p* = 0.02, 95%CI: 1.175–6.551) and the use of freehand needle-knife precut technique (OR 5.203, *p* = 0.006, 95%CI: 1.612–16.795) was also linked to an increased risk for PEP in multivariate analysis.

Difficult cannulation (OR 4.270, *p* < 0.001, 95%CI: 1.940–5.397), and incomplete sphincterotomy (OR 3.976, *p* = 0.032, 95%CI: 1.122–4.086) were identified as individual predictive risk factors for post-ERCP bleeding, while balloon sphincteroplasty (OR 0.279, *p* = 0.043, 95%CI: 0.181–0.961) seemed to be a protective factor. In multivariate analysis, insertion of biliary stents was a risk factor for post-ERCP infections (OR 0.425, *p* = 0.036, 95%CI: 0.177–0.823), while complete biliary sphincterotomy had a protective role (OR 2.467, *p* = 0.026, 95%CI: 1.114–5.463).

## 4. Discussion

Most of the experienced endoscopists can differentiate among several characteristics of the papilla but, to date, the impact of such endoscopic findings on ERCP outcomes and adverse events rates have not been clearly defined. Only limited research that focused mainly on the impact of papillary morphology on deep cannulation could be identified [11,15,16]. One of the main reasons for such paucity within the literature is that native papillae have such variable appearances that a commonly accepted classification does not yet exist [12]. In this regard, the main contribution within the literature of the last decade comes from a team of researchers from Scandinavia that published the first interobserver- and intraobserver-validated classification of the papillary morphology [15]. The same team has recently published an original research on the role of the different types of papilla in bile duct cannulation, proving that papillary morphology affects bile duct cannulation [16].

Despite its multicenter validation and systematic description of morphology for the four different types of papilla, the classification scheme was described as limited, on one hand due to the non-inclusion of some other relatively frequent papilla types and, on the other hand, because of the lack of evidence regarding the risk that such anatomical variations pose on the post-ERCP occurrence of adverse events [12]. Therefore, the expanded systematic arrangement in our study aimed to cover some other papillary aspects like the small and retracted papillae or the large, hooded papillae with multiple folds over the orifice. Furthermore, our study went a step further and evaluated not only the impact on cannulation, but also that on ERCP outcomes and adverse events. Nevertheless, by covering supplementary endoscopic characteristics of the papilla, we needed to consider the limitation of using a non-validated classification. However, suggestions for such classifications were made within several publications [10,17]. While using an interobserver- and intraobserver-validated classification proves useful and relevant in an everyday clinical setting [15,16], adding up to such classification would not provide less generalizable results within a single center prospective cohort.

As in the previous multicenter studies, regular papillae are the ones most frequently encountered also in our cohort [15,16]. In what the impact of anatomy on cannulation is concerned, the study demonstrated that Type 1 small and/or retracted papillae are more frequently difficult to cannulate compared to the regular ones, and the results are consistent with those of the recent multicenter study from Haraldsson et al. [16]. Similar findings that have been reported previously only by expert opinion [11,19,20] are now being validated by prospective studies. Moreover, our study shows non-homogenous usage for the alternative rescue papillotomy techniques in cases which are difficult to cannulate. This should be correlated with the fact that, as advised recently by expert opinions in the field [12], our classification took into account the length of the intraduodenal portion of the common bile duct that seems to impact the choice of alternative access techniques.

The endoscopic appearance of the papilla was a risk factor for both PEP and overall post-ERCP risk for adverse events in univariate analysis. Moreover, it was proven by the AUROC analysis to be predictive for difficult cannulation. Such correlation has not been confirmed by previous studies [13,16]. Given the fact that in the present study difficult cannulation has been itself proven an individual and predictive risk factor for PEP, post-procedural bleeding and for the overall post-ERCP risk for adverse events in multivariate analysis, the morphology of major papilla should play at least an indirect role. The higher rates of PEP associated with difficult cannulation are by now well described and thoroughly documented by both prospective studies, meta-analyses [21,22,23] and guideline publications [4,6,8,24,25]. Otherwise, its correlation proved within our cohort with post-ERCP bleeding and with an overall elevated risk for post-ERCP adverse events seems to be novel. The rates of failed cannulation and PEP in the study were consistent with those cited in the current guidelines [6,8,25].

Interestingly, Type 4 papillae were individual and predictive risk factors for PEP, contrary to previous expert opinion that only small and/or retracted papillae could add on such risk [10,11]. To our knowledge there is no similar study confirming such assumption. Nevertheless, Haraldsson et al. found that the protruding or pendulous papillae are often more difficult to cannulate [16], thus demonstrating indirectly the possible correlation. The possible contributive factors for such association within the analyzed cohort were represented by increased frequency of rescue needle-knife papillotomy techniques (fistulotomy or freehand precut) and prophylactic pancreatic stenting for patients with Type 4 papillae. Nevertheless, the frequency of pancreatic stenting was low mainly due to the fact that prophylactic pancreatic stenting was performed, as recommended by current guidelines, only in high-risk patients [26] with easy pancreatic access (as defined by Dumonceau et al., 2020) and contraindication to high-volume hydration [25,26]. Therefore, such patients may have been exposed to inadvertent pancreatic duct manipulation that itself has been proven an individual and predictive risk factor for PEP [23].

The present study was not designed to investigate the influence of endoscopists’ experience on successful cannulation of the various types of papilla and subsequent ERCP outcomes. Such limitation came from the fact that within our medical center there were only two senior advanced endoscopy fellows with an experience in ERCP allowing them to practice. An experienced endoscopist intervened only in unsuccessful cases. Haraldsson et al. found that when a fellow endoscopist starts the ERCP, the rate of failed cannulation rises, although independently, on any type of papillary morphology [16]. Further dedicated studies on training in ERCP that would consider the impact of duodenal and papillary anatomy on cannulation and outcomes are clearly required.

## 5. Conclusions

The present study described an expanded classification of the endoscopic appearance of the major papilla covering most of the papillary morphologies. The various types of papilla have been significantly correlated to different rates of difficult cannulation, small and/or retracted papillae being more frequently difficult to cannulate. After thorough prospective monitoring, it has been shown that the anatomy of the papilla may pose a different risk on the overall post-ERCP adverse events rates, and subsequently, large and folded papillae can be regarded to as independent and predictive risk factors for PEP as such patients may be prone to inadvertent pancreatic duct manipulation and pancreatic stenting. Such information can help with decision making and implementation of rescue techniques during the procedure, but should be further validated by multicenter prospective trials.

## Figures and Tables

**Figure 1 jcm-09-01637-f001:**
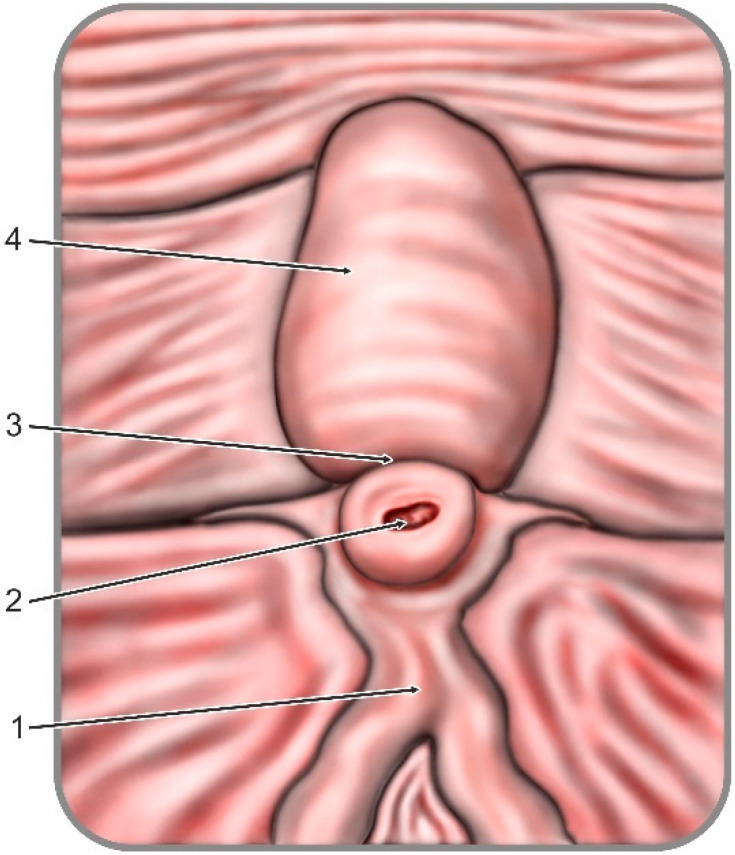
Regular anatomy of the papilla: (1) frenulum; (2) orifice; (3) recessus; and (4) infundibulum. Adapted after Canard et al., 2011 [17].

**Figure 2 jcm-09-01637-f002:**
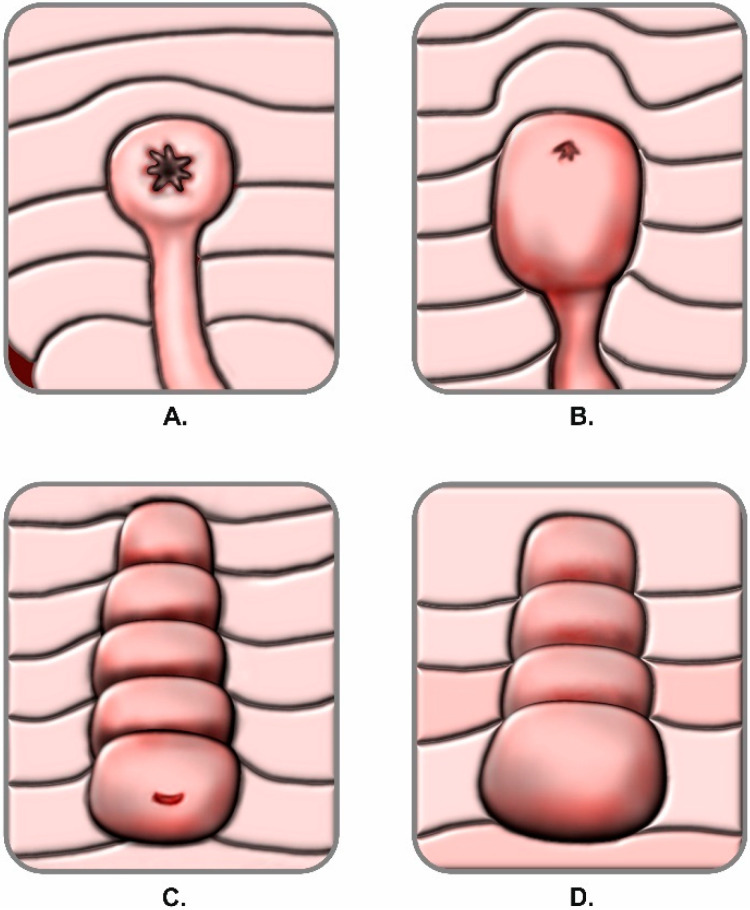
Anatomical variations of the papilla: (**A**) Type 1; (**B**) Type II; (**C**) Type III; (**D**) Type IV. Adapted after Canard et al., 2011 [17].

**Figure 3 jcm-09-01637-f003:**
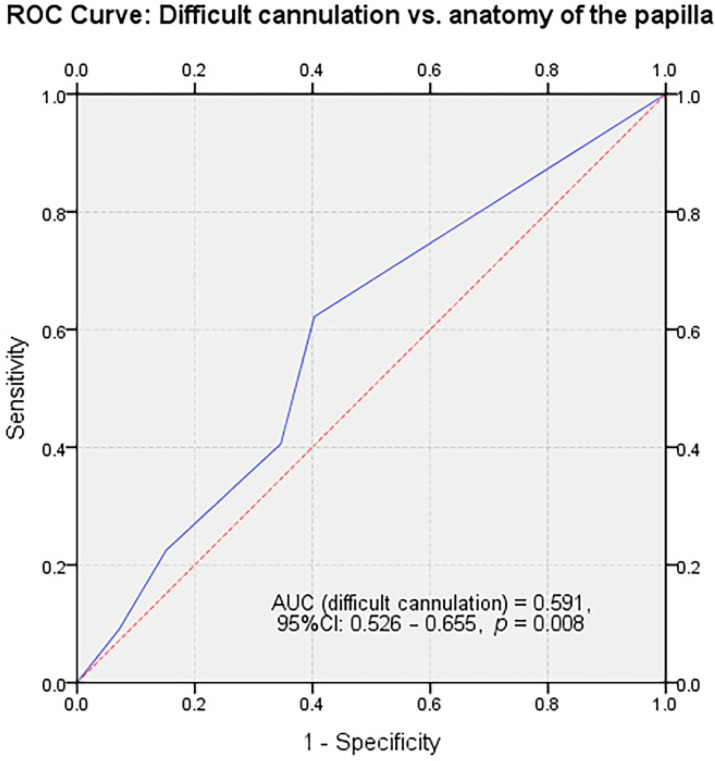
Receiver operating characteristic (ROC) curve for evaluating the predictive power of different papilla types on the rate of difficult cannulation.

**Figure 4 jcm-09-01637-f004:**
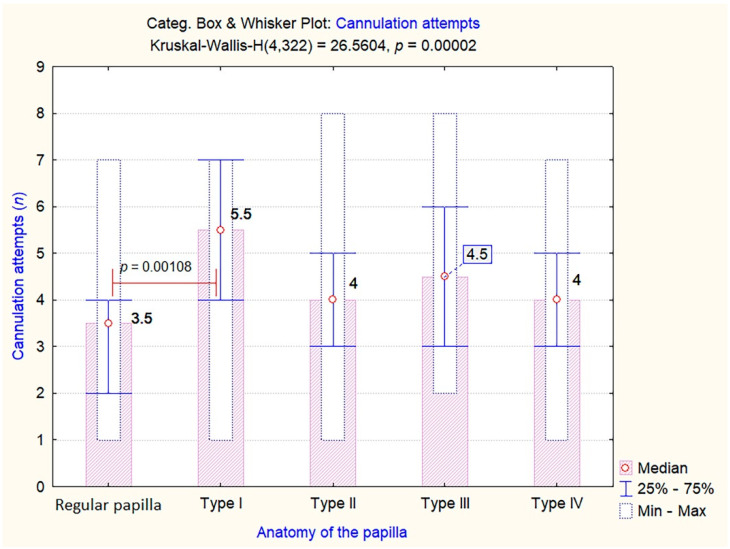
Number of cannulation attempts among the different endoscopic aspects of the papilla. Levene Test of Homogeneity of Variances: F = 1.3557, *p* = 0.249188.

**Figure 5 jcm-09-01637-f005:**
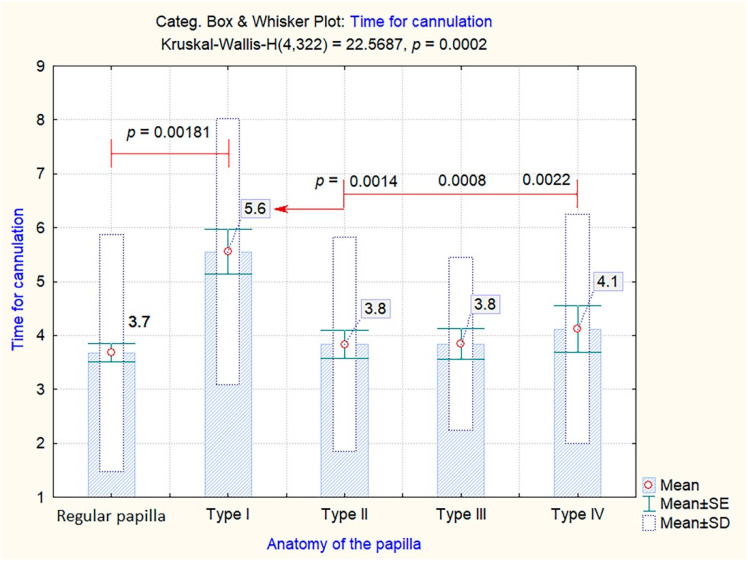
Time for successful cannulation among the different endoscopic aspects of the papilla Levene Test of Homogeneity of Variances: F = 0.603396, *p* = 0.660458.

**Table 1 jcm-09-01637-t001:** Distribution of the different endoscopic aspects of the papilla.

Papilla	*n* (%)
Regular	168 (52.1%)
Type 1	36 (11.1%)
Type 2	61 (18.9%)
Type 3	32 (9.9%)
Type 4	25 (7.7%)

**Table 2 jcm-09-01637-t002:** Difficult cannulation among different papilla types.

Papilla	Cannulation				Total	95% CI	
	Standard		Difficult			Yates Chi-Square = 24.96	
Regular	126	75%	42	25%	168	r = 0.282	*p* = 0.00358
Type 1	12	33.3%	24	66.7%	36		
Type 2	41	67.2%	20	32.8%	61		
Type 3	17	53.1%	15	46.9%	32		
Type 4	15	60%	10	40%	25		

**Table 3 jcm-09-01637-t003:** Univariate analysis showing correlations between the different endoscopic appearances of the papilla and post-Endoscopic retrograde cholangiopancreatography (ERCP) adverse events rates.

Papilla (*n* = 322)	Overall Post-ERCP Adverse Events		Test ^†^	*p*-Value * (95%CI)
	Absent (*n* = 262)	Present (*n* = 60)		
Regular papilla	140 (83.33%)	28 (16.67%)	16.087	0.0066 *
Type 1	26 (72.22%)	10 (27.78%)		
Type 2	53 (86.89%)	8 (13.11%)		
Type 3	29 (90.63%)	3 (9.38%)		
Type 4	14 (56%)	11 (44%)		
	Post-ERCP pancreatitis			
	Absent (*n* = 289)	Present (*n* = 33)		
Regular papilla	150 (89.29%)	18 (10.71%)	13.275	0.01001 *
Type 1	31 (86.11%)	5 (13.89%)		
Type 2	59 (96.72%)	2 (3.28%)		
Type 3	31 (96.88%)	1 (3.13%)		
Type 4	18 (72%)	7 (28%)		
	Post-ERCP bleeding			
	Absent (*n* = 312)	Present (*n* = 10)		
Regular papilla	163 (97.02%)	5 (2.98%)	2.9099	0.5730
Type 1	36 (100%)	0 (0%)		
Type 2	58 (95.08%)	3 (4.92%)		
Type 3	31 (96.88%)	1 (3.13%)		
Type 4	24 (96%)	1 (4%)		
	Post-ERCP infections			
	Absent (*n* = 299)	Present (*n* = 23)		
Regular Papilla	160 (95.24%)	8 (4.76%)	9.145977	0.05756
Type 1	29 (80.56%)	7 (19.44%)		
Type 2	57 (93.44%)	4 (6.56%)		
Type 3	31 (96.88%)	1 (3.13%)		
Type 4	22 (88%)	3 (12%)		

Variables: *n* (%). ^†^ Pearson Chi-square test; * Marked effects are significant at *p* < 0.05.

**Table 4 jcm-09-01637-t004:** Correlations between the different alternative papillotomy techniques and the endoscopic appearances of the papilla.

Papilla	Rescue Papillotomy Techniques						Total	95% CI	
	Needle-Knife Freehand Precut		Needle-Knife Fistulotomy		Transpancreatic Biliary Sphincterotomy			M-L Chi-Square = 32.6658	
Regular	7	4.17%	-	-	3	1.79%	168	r = 0.3777	*p* = 0.01475
Type 1	2	5.56%	-	-	2	5.56%	36		
Type 2	8	13.11%	-	-	2	3.28%	61		
Type 3	3	9.38%	3	9.38%	2	6.25%	32		
Type 4	2	8%	1	4%	-	-	25		

**Table 5 jcm-09-01637-t005:** Distribution of pancreatic stent insertion among different types of papilla.

Papilla	Prophylactic Pancreatic Stent		95% CI	
			Yates-Chi-Square = 1.60275	
Regular	5	2.98%	r = 0.2119129	*p* = 0.75415
Type 1	1	2.78%		
Type 2	3	4.92%		
Type 3	1	3.13%		
Type 4	2	8.00%		

**Table 6 jcm-09-01637-t006:** Multiple regression. Identification of procedure-related risk factors for post-ERCP adverse events.

Multiple Regression	SE	Wald Test	P	Odd Ratio	95% CI	
					Lower	Upper
Overall post-ERCP adverse events						
Papillary morphology (ref.: regular papilla)		7.324	0.198			
Type 1	0.515	0.985	0.321	0.600	0.219	1.646
Type 2	0.546	1.001	0.317	0.579	0.199	1.688
Type 3	0.458	2.982	0.084	0.453	0.185	1.113
Type 4	0.507	0.552	0.458	1.457	0.540	3.932
Constant	0.430	2.270	0.132	0.523		
Duodenal diverticulum (ref.: absent diverticulum)		1.487	0.685			
Type 1	0.272	0.715	0.999	0.231	0.142	0.528.
Type 2	0.464	0.684	0.408	1.468	0.591	3.642
Type 3	0.080	0.716	0.397	0.401	0.048	3.331
Difficult cannulation	0.299	11.370	0.001 *	2.744	1.526	4.933
Alternative access papillotomy (ref.: standard biliary sphincterotomy)		2.472	0.480			
Needle-knife freehand precut	0.473	0.543	0.461	1.417	0.561	3.581
Needle-knife fistulotomy	0.079	0.370	0.543	1.929	0.233	15.994
Transpancreatic biliary sphincterotomy	0.834	1.303	0.254	0.386	0.075	1.979
Altered biliary anatomy (ref.: normal anatomy)	0.379	0.286	0.593	0.817	0.389	1.716
Bile duct stones (ref.: absence of stones)	0.375	3.107	0.078	0.516	0.247	1.077
Post-ERCP pancreatitis						
Papillary anatomy (ref.: normal papilla)		15.453	0.009 *			
Type 1	0.921	1.939	0.164	3.605	0.593	21.924
Type 2	0.283	0.034	0.854	0.789	0.064	9.762
Type 3	0.055	0.107	0.744	0.708	0.089	5.603
Type 4	0.889	7.901	0.005 *	12.176	2.131	69.567
Difficult cannulation	0.438	5.421	0.020 *	2.775	1.175	6.551
Alternative access papillotomy (ref.: standard biliary sphincterotomy)		7.804	0.050 *			
Needle-knife freehand precut	0.598	7.610	0.006 *	5.203	1.612	16.795
Needle-knife fistulotomy	0.934	0.000	0.999	0.000	0.000	0.001
Transpancreatic biliary sphincterotomy	0.130	0.000	0.995	1.007	0.110	9.219
Constant	0.744	29.358	0.000	0.018		
Post-ERCP bleeding						
Difficult cannulation	0.402	13.012	<0.001 *	4.270	1.940	5.397
Pancreatic duct cannulation	0.417	0.687	0.407	1.413	0.624	3.200
Papillotomy (ref.: no papillotomy)		4.577	0.03 1*			
Complete biliary sphincterotomy	0.665	3.319	0.068	3.356	0.912	12.346
Incomplete papillotomy	0.645	4.573	0.032 *	3.976	1.122	4.086
Brush cytology: malignant	0.735	6.580	0.010 *	6.592	1.560	7.845
Balloon sphincteroplasty	0.630	4.096	0.043 *	0.279	0.181	0.961
Constant	0.635	38.612	<0.001	0.019		
**Post-ERCP infections**						
Papillotomy (ref.: no papillotomy)		4.336	0.114			
Complete biliary sphincterotomy	0.448	3.645	0.036 *	0.425	0.177	0.823
Incomplete papillotomy	0.528	2.626	0.010 *	0.428	0.151	0.896
Indication: bile duct stones (ref.: stenoses)	0.475	0.781	0.377	0.657	0.259	1.667
Biliary stent insertion (ref.: absence of stents)	0.406	4.956	0.026 *	2.467	1.114	5.463
Constant	0.337	28.388	0.000	0.166		

* Marked effects are significant at *p* < 0.05. CI—confidence interval, SE—standard error.

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
