# Peer review of "Anatomy of Major Duodenal Papilla Influences ERCP Outcomes and Complication Rates: A Single Center Prospective Study"

_jcm, 2020, doi:10.3390/jcm9061637_

Round 1
Reviewer 1 Report
In this study authors evaluated relationship between papilla morphology and ERCP outcomes, in particular rates of difficult cannulation and complications.
The strength of the study is the analysis of various aspects of difficult cannulation such as number of cannulation attempts, cannulation times, advanced cannulation techniques utilized and pancreatic stenting. The main weakness of the study is the use of a papilla morphology classification which is not widely accepted or validated. The classification system studied is described by authors as broad but is only slightly different from previously published. The effect of papilla morphology on cannulation was evaluated in a recently published multicenter trial. The study under review does not sufficiently build on the existing publication.
Major comments
Increasing number of patients studied (patients were included only over 8-month period) would strengthen the study. Increasing number of patients might also allow authors to look into how cannulation approach by expert endoscopist is influenced by papilla morphology, i.e. early use of advanced cannulation techniques such as double wire or precut for small or large/protruding papilla. Authors should also consider re-evaluating the data using previously published papilla morphology classification which would allow for broader application of findings.
Minor comments
- Methods: Was there specific post-procedure hydration protocol used?
- Methods: Please clarify if follow up at 15 and 60 days was in-person or over phone follow-up
- Methods: post-procedural mortality was recorded. Please clarify if there were deaths after ERCP.
- Table 1: # do not add up to 322. Please clarify
- Table 2: # of type 4 papilla does not add up to # for type 4 in Table 1. Please clarify
- In Fig 4, 5 is "normal" papilla the same as regular papilla. Please clarify
- English Grammar editing, in particular Results lines 209-216
Author Response
Dear Reviewer 1,
We have uploaded a response letter where we tried to explain how we managed to cover up the most of your advice and questions. Hopefully we have reached a better version of our manuscript. Thank you for you assessment and feed-back!
Sincerely,
Gheorghe G. Balan on behalf of all authors.

Reviewer 2 Report
I read with great interest your manuscript about the impact of the papilla morphology in ERCP cannulation rate and post-ERCP complications.
The topic is original, as there are few papers published about this topic.
However I have one major question that I would like to hear your opinion.
You use a papilla classification developed by your group, that, based in what is described in the methods section, was not properlly validated. Moreover it's not described the process undertaken to build the classification: Delphi method, other methodology; criteria to define the various papilla categories, etc
Why did you not used a validated classification, or performed a proper validation of your classification, before embarking in this study.
Author Response
Dear Reviewer 2,
We have uploaded a response letter where we tried to explain how we managed to cover up your advice and questions. In order to provide clear explanations, we have separated your feedback into multiple sections and we answered to each of them to the best of our capacity. Hopefully we have reached a better version of our manuscript. Thank you for you assessment!
Sincerely,
Gheorghe G. Balan on behalf of all authors.

Round 2
Reviewer 1 Report
I appreciate authors efforts to improve the study, and most of my concerns have been answered. I understand that due to local procedural protocols it is not possible at this time to fully evaluate cannulation patterns based on papilla morphology.
Based on authors responses it appears that the main goal of the study was to expand upon the existing papilla morphology classification. The manuscript in its current form does not make this clear. I wold recommend revising last paragraph of the introduction to clearly state that the goal of the study is evaluation of the expanded papilla morphology classification rather than development of the alternative classification. There is published literature on this subject thus I would also consider avoiding statements about "lack of studies".
Author Response
Dear Reviewer 1,
We hope to have answered once again your concerns and advice in a suitable manner. Please see the detailed attachment.
Best regards,
Thankfully,
Gheorghe G. Balan
